 

# Gliogenesis from the subventricular zone modulates the extracellular matrix at the glial scar after brain ischemia

Maria Ardaya[1,2,3†], Marie-Catherine Tiveron[4], Harold Cremer[4], Esther Rubio-López[1,5], Abraham Martín[1,6], Benjamin Dehay[7], Fernando Pérez-Cerdá[1,2,3], Carlos Matute[1,2,3], Federico N Soria[1,2,3,6]*, Fabio Cavaliere[1,2,3,8]*

[1]Achucarro Basque Center for Neuroscience, Leioa, Spain; [2]Department of Neuroscience, University of the Basque Country (UPV/EHU), Leioa, Spain; [3]CIBERNED, Madrid, Spain; [4]Aix Marseille University, CNRS, IBDM, Campus de Luminy, Marseille, France; [5]CIC biomaGUNE, Basque Research and Technology Alliance (BRTA), San Sebastian, Spain; [6]Ikerbasque, Basque Foundation for Science, Bilbao, Spain; [7]University of Bordeaux, CNRS, IMN, UMR 5293, Bordeaux, France; [8]Basque Biomodel Platform for Human Research (BBioH), Leioa, Spain

**\*For correspondence:**
federico.soria@achucarro.org
(FNS);
fabio.cavaliere@ehu.eus (FC)

**Present address:** †Donostia
International Physics Center
(DIPC), San Sebastian, Spain

**Competing interest:** The authors
declare that no competing
interests exist.

**Reviewing Editor:** Annalisa
Scimemi, University at Albany,
State University of New York,
Albany, United States

## eLife Assessment

The authors show that a middle carotid artery occlusion (MCAO) hypoxia lesion leads to hyaluronan-mediated chemoattraction to the lesion penumbra of Thbs-4-expressing astrocytes of the subventricular zone (SVZ). These findings are **valuable** because they shed light on the function of astrocytes from the adult SVZ in pathological states like brain ischemic injury. The results are **convincing** as they rely on a comprehensive analysis of experimental data.

**Abstract** Activation of the subventricular zone (SVZ) following cerebral ischemia is one of the brain's early responses to counteract neuron loss and minimize tissue damage. Impaired brain regions communicate with the SVZ through various chemotactic signals that promote cell migration and differentiation, primarily involving neural stem cells, neuroblasts, or glioblasts. However, the activation of gliogenesis and the role of newly formed astrocytes in the post-ischemic scenario remain subjects of debate. We have previously demonstrated that adenosine release after brain ischemia prompts the SVZ to generate new astrocytes. Here, we used transient brain ischemia in mice to identify the cellular origin of these astrocytes within the SVZ neurogenic niche and investigate their role in the pathological process. By combining immunofluorescence, BrdU-tracing, and genetic cell labeling, we tracked the migration of newborn astrocytes, positive for the proteoglycan marker Thbs4, from the dorsal and medial SVZ to the perilesional barrier surrounding the ischemic core, known as the 'glial scar'. We found that these Thbs4-positive astrocytes modulate the dense extracellular matrix at the lesion border by both synthesizing and degrading hyaluronan. We also show that while the accumulation of hyaluronan at the lesion site is sufficient to recruit newborn astrocytes, its degradation at the SVZ correlates with gliogenesis. These findings suggest that newborn astrocytes could be a promising pharmacological target for modulating the glial scar after brain ischemia and facilitating tissue regeneration.

## Introduction

Cell regeneration in the adult mammalian brain primarily occurs in two neurogenic niches: the subventricular zone (SVZ) in the lateral ventricle and the subgranular zone (SGZ) in the dentate gyrus of the hippocampus (*Bond et al., 2015*). Adult brain stem cells are multipotent and originate from the embryonic radial glia, thus sharing key cell markers and features with resident astrocytes. In the adult mouse brain, neuroblasts from SVZ migrate through the rostral migratory stream (RMS) to the olfactory bulb (OB), where neurogenesis takes place (*Luskin, 1993*). Neuronal hyperactivity (e.g., during epilepsy), as well as pathological conditions (e.g., stroke) or natural (e.g., aging) processes, can impair the physiological functions of the neurogenic niches, altering their neurogenic and gliogenic potential (*Chaker et al., 2016*; *Kralic et al., 2005*; *Sundholm-Peters et al., 2005*). For example, SVZ cells are known to become activated in vivo following focal ischemia induced by middle cerebral artery occlusion (MCAO) (*Ceanga et al., 2021*).

We have previously demonstrated in oxygen and glucose-deprived (OGD) organotypic brain slices that the ischemic region and the SVZ establish a biochemical interplay based on a gradient of toxic and protective factors released early after the insult (*Cavaliere et al., 2006*). Furthermore, others have demonstrated that brain stroke models in mice can activate astrogliogenesis at the expense of neurogenesis (*Benner et al., 2013*; *Laug et al., 2019*; *Li et al., 2010*). Benner and colleagues identified a SVZ-derived astrocyte population characterized by high levels of the membrane-bound proteoglycan thrombospondin 4 (Thbs4), a cell–cell and cell–matrix adhesion molecule that has been considered neuroprotective against ischemic damage (*Benner et al., 2013*; *Laug et al., 2019*). Ablation of Thbs4 in KO mice also produced abnormal ischemic scar formation and increased microvascular hemorrhage, suggesting a role of SVZ astrogliogenesis in brain damage and glial scar formation induced by ischemic stroke (*Benner et al., 2013*).

The perilesional barrier known as the *glial scar* assembles early after brain ischemia, isolating the damaged area and limiting both the spread of pathogens and potentially toxic damage-associated molecular patterns (DAMPs) into the surrounding area (*Anderson et al., 2016*; *Bush et al., 1999*; *Faulkner et al., 2004*). Consisting of packed reactive astrocytes and a dense extracellular matrix (ECM) composed mostly of hyaluronic acid (HA), the glial scar represents a physical barrier for the SVZ-generated newborn neurons that migrate into the ischemic core (*Arvidsson et al., 2002*; *Yamashita et al., 2006*). Thus, modulating glial scar formation after brain ischemia may be a strategy to reduce tissue damage and restore brain functionality. In ischemic stroke, the brain ECM goes through a significant reorganization due to injury and must undergo successful remodeling to promote brain repair. ECM elements involved in the ischemic cascade induce activation of astrocytes and microglia, which modulate ECM structure by releasing glycoproteins into the extracellular space (*Adams and Gallo, 2018*; *Fawcett, 2015*). This crosstalk culminates in the formation of the glial scar, which is created locally by reactive astrocytes and astrocytes-secreted ECM molecules such as HA and chondroitin sulphate proteoglycans (*Bush et al., 1999*; *Lau et al., 2013*).

We have previously demonstrated that high extracellular adenosine levels, a hallmark of ischemic insults, induce astrogliogenesis from the SVZ (*Benito-Muñoz et al., 2016*). Following our previous results, here we characterize the SVZ response to brain ischemia and investigate the role of newly generated astrocytes in the modulation of the glial scar after MCAO. We show that ischemic conditions increase the number of newly generated astrocytes in the SVZ and their migration to the ischemic area. We also propose the role of these astrocytes in modulating the ECM at the glial scar and suggest this astrocyte population as a possible target for brain repair after brain ischemia.

## Results

### Thbs4 is expressed in the neurogenic niche of the SVZ

As suggested by others (*Benner et al., 2013*; *Girard et al., 2014*; *Pous et al., 2020*), Thbs4 is highly expressed in astrocytes of the SVZ and RMS and might play a role in ischemia-induced neurogenesis. To confirm these observations, we labeled the mouse SVZ by immunofluorescence for Thbs4. Thbs4 was mainly expressed in the SVZ, RMS, and OB but also in the cerebellum and ventral tegmental area (*Figure 1A*). In the SVZ, the staining was associated with cells with different morphologies (*Figure 1B*), ranging from cells with small primary apical processes in contact with the ventricle, to rounded NSC-like cells in the dorsolateral SVZ horn. After quantification by direct count of Thbs4-positive cells,



**Figure 1.** Thbs4 labels type-B cells in the subventricular zone (SVZ) neurogenic niche. (**A**) Schematic drawing of the brain showing Thbs4 labeling throughout the mouse brain. Thbs4 was primarily found in the SVZ and rostral migratory stream (RMS) of the adult mouse telencephalon. Arrows indicate nonspecific Thbs4 staining in the cerebellum and ventral tegmental area. (**B**) Thbs4-positive cells in the SVZ displayed various morphologies. (**C**) Proportion of Thbs4-positive cells relative to the total neural stem cells (NSCs) population in the SVZ. (**D**) Classification of Thbs4-positive cells in the SVZ based on their NSC morphology. (**E**) Electroporation of pThbs4-GFP at P1 revealed Thbs4 expression in the postnatal dorsal SVZ. (**F**) Quantification of Thbs4-reporter expression showing the proportion of NSC expressing Thbs4 postnatally in the dorsal SVZ. (**G**) Thbs4-GFP reporter expression was

*Figure 1 continued on next page*

*Figure 1 continued*

upregulated in the rostral SVZ. BV: blood vessel; LV: lateral ventricle. Scale bar = 10 µm (**B**) and 50 µm (**E**). Bars represent mean ± SEM. *p<0.05 and **p<0.01, Tukey post hoc test (after one-way ANOVA was significant at p<0.05).

Thbs4-positive astrocytes represented 21.4% of the total SVZ cells (*Figure 1C*), with more than 60% of them displaying a radial glia-like type-B morphology (*Bond et al., 2015*; *Doetsch et al., 1999*; *Figure 1D*). This is in accordance with other studies reporting Thbs4 expression in B-cells, either by immunohistochemistry (*Beckervordersandforth et al., 2010*; *Benner et al., 2013*) or single-cell RNA sequencing (*Basak et al., 2018*; *Cebrian-Silla et al., 2021*; *Llorens-Bobadilla et al., 2015*).

To validate the immunohistochemistry protocol for Thbs4, we labeled the SVZ cells with green fluorescent protein (GFP) under the control of the Thbs4 promoter (pThbs4-eGFP) by postnatal (P1) electroporation (*Figure 1E*). Twenty-four hours after electroporation, almost 20% of total cells in the dorsal SVZ expressed GFP (*Figure 1F*), with a greater expression in the rostral SVZ (*Figure 1G*), confirming that Thbs4-positive cells were mainly localized in the neurogenic niche of the SVZ. This also suggests that Thbs4 is expressed in NSCs at early postnatal stages, generating astrocytes during development.

## Thbs4 expression in the SVZ increases after brain ischemia

To investigate whether brain ischemia could activate astrogliogenesis in the SVZ, we analyzed cell proliferation and Thbs4 expression in the SVZ after MCAO (*Figure 2*). Sixty minutes of MCAO (*Figure 2—figure supplement 1A*) caused neuronal damage in the cortex and striatum, as observed by cresyl violet staining and NeuN immunofluorescence (*Figure 2—figure supplement 1B and C*). The survival rate at 28 days post-lesion was around 50% (*Figure 2—figure supplement 1D*). Moreover, motor deficits and animal weight of ischemic mice were monitored every day after MCAO for 30 days. Motor deficits improved over time, while body weight dropped sharply the first 3 days after MCAO and was partially restored at 30 dpi (*Figure 2—figure supplement 1E and F*).

To study cell proliferation in the SVZ, we administered 50 mg/kg BrdU i.p. every 2 hours (three doses in total) the day before MCAO (*Figure 2—figure supplement 2A*). We observed an increase in the number of total SVZ BrdU-positive cells 24 hours after MCAO (*Figure 2—figure supplement 2B and C*). Ki67 immunofluorescence of SVZ wholemount preparations also revealed a more than tenfold increase of proliferative cells 24 hours after MCAO (*Figure 2—figure supplement 2D and E*). In addition, we observed a significant increase in cleaved Caspase3-positive cells 24 hours post-lesion (*Figure 2—figure supplement 2F and G*). These results suggest a fast and specific proliferative response in the ischemia-induced SVZ.

We next analyzed Thbs4 staining in combination with Nestin, an NSC marker, and doublecortin (DCX), a marker of neuroblasts, before and after MCAO. The entire SVZ was analyzed in sham and ischemic mice 7, 15, and 30 days post-ischemia (dpi). The immunofluorescence analysis of the SVZ (*Figure 2A*) showed a transient decrease in Nestin and DCX-positive cells but an increase of Thbs4-positive astrocytes up to 15 dpi (*Figure 2B–D*) compared to sham animals. The increase of Thbs4 astrocytes was also confirmed by western blot analysis using SVZ homogenates (*Figure 2E and F*), evidencing a twofold increased expression of Thbs4 in the ischemic SVZ. Similarly to immunofluorescence, western blot also showed decreased Nestin protein levels at 15 dpi (*Figure 2G*). These results demonstrate that Thbs4 expression in the SVZ increases after ischemia, coinciding with a decrease in neuroblast markers, suggesting that the classical pro-neuroblast program of the SVZ is interrupted in favor of the generation of astrocytes after ischemia.

## Characterization of ischemia-induced cell populations in the SVZ

To characterize the different cell populations of the SVZ activated after focal ischemia, we labeled the whole pool of proliferating cells with 1% BrdU in drinking water for 14 days before MCAO (*Codega et al., 2014*). One month after MCAO, BrdU in the SVZ primarily stained slow proliferative cells like type B cells, whereas in rapid transit-amplifying cells (type C), BrdU was diluted away. Therefore, we chose to label type C cells by intraperitoneal administration (50 mg/kg) of 5-Iodo-2'-deoxyuridine (IdU) 24 hours before euthanasia (*Figure 3A*). To identify the cellular origin of newborn astrocytes, we analyzed the SVZ 30 dpi by immunofluorescence co-labeling Thbs4 and GFAP together with BrdU or IdU. We found a twofold increase in the total amount of proliferative cells (BrdU-positive) after MCAO with respect to the sham group (*Figure 3B*), whereas we did not observe a significant change



**Figure 2.** Thbs4 expression is upregulated in the subventricular zone (SVZ) after brain ischemia. (**A**) Representative images of Thbs4 (green), Nestin (red), and doublecortin (DCX) (magenta) in the SVZ under physiological conditions (left) and 15 days post-injury (dpi) (right). (**B, C**) The number of actively proliferating neural stem cells (NSCs) and neuroblasts decreases at 7 and 15 dpi in the SVZ, as shown by reduced Nestin (**B**) and DCX (**C**) levels, respectively. (**D**) Thbs4 expression increases in the SVZ over time following ischemic injury. (**E**) Representative western blot images of Nestin and Thbs4 in sham and 15 dpi SVZ. (**F, G**) Quantification of mean gray value (MGV) from (**E**), normalized to GAPDH, shows an increase in Thbs4 (**F**) and a decrease in Nestin (**G**) in the 15 dpi SVZ. Scale bar = 100 μm in (**A**). n=6 (**B–D**) and 3 (**E–G**) per condition. Bars represent mean ± SEM. *p<0.05 and **p<0.01; two-tailed Student's *t*-test (sham vs. 15ddMCAO) and Tukey post hoc test (after one-way ANOVA was significant at p<0.05).

The online version of this article includes the following source data and figure supplement(s) for figure 2:

*Figure 2 continued on next page*

*Figure 2 continued*

**Source data 1.** Full uncut blots (annotated) for *Figure 2E*.

**Source data 2.** Raw uncut blot files.

**Figure supplement 1.** Characterization of the middle cerebral artery occlusion (MCAO) mouse model of brain ischemia.

**Figure supplement 2.** Cell proliferation in the subventricular zone (SVZ) after brain ischemia.

in the number of IdU-positive cells (*Figure 3C*). The number of slow proliferative NSC (Thbs4/GFAP/BrdU) increased linearly after MCAO (*Figure 3D and E*), especially in the dorsal region of the SVZ (*Figure 3—figure supplement 1A*), whereas the number of activated NSC (Thbs4/GFAP/IdU) did not change significantly, either in the whole SVZ (*Figure 3F*) or in any subregion analyzed (*Figure 3—figure supplement 1*). These results suggest that, after ischemia, Thbs4-positive astrocytes derive from the slow proliferative type B cells.

## Ischemia-induced newborn astrocytes migrate from the dorsal SVZ to ischemic regions

To confirm that Thbs4-positive astrocytes originate from type B cells of dorsal SVZ, we labeled NSCs of dorsal SVZ by electroporating the pCAGGSx-CRE plasmid in P1 Ai14 Rosa26-CAG-tdTomato transgenic mice (*Figure 4A*). Plasmid electroporation induced the expression of tdTomato (tdTOM) by Cre recombination, enabling the spatial tracking and cell fate assessment of electroporated NCSs from

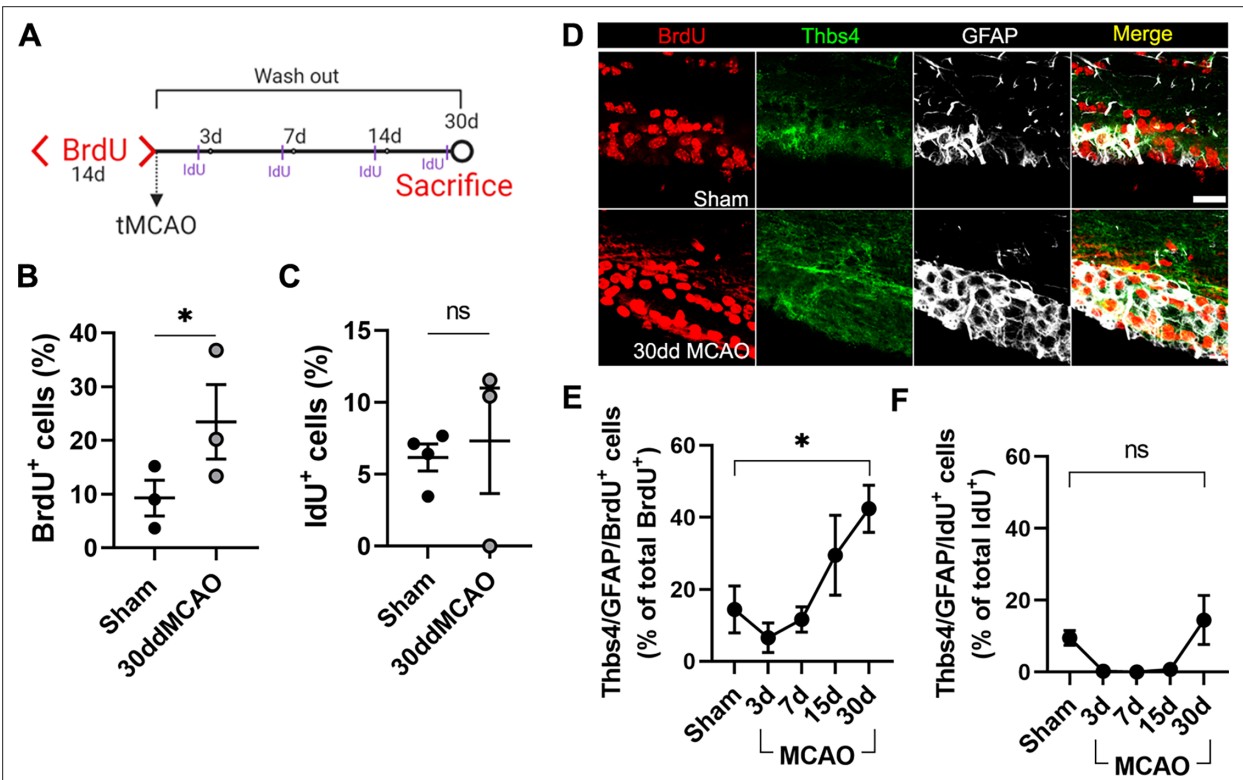

**Figure 3.** Thbs4-positive cells derive from type B neural stem cells (NSCs) in the subventricular zone (SVZ) after brain ischemia. (**A**) Experimental design: chronic BrdU (1% in water) treatment was administered for 2 weeks. BrdU labeling persisted only in B type NSCs 30 days after treatment. Animals were injected with IdU (50 mg/kg) three times the day before euthanasia to label proliferating cells. (**B, C**) BrdU (**B**) and IdU (**C**)-positive cells in the SVZ. Only BrdU-positive cells showed a significant increase at 30 dpi. (**D**) Representative confocal images showing Thbs4, GFAP, and BrdU levels in sham (top) and 30 dpi mice (bottom). (**E, F**) Thbs4/GFAP/BrdU-positive cells (slow proliferative cells) increased in the SVZ at 30 dpi (**E**), while no changes were observed in Thbs4/GFAP/IdU-positive cells (fast proliferative cells). Scale bar = 20 µm (**D**). n=4 (sham) and 3 (MCAO). Bars represent mean ± SEM. *p<0.05 by two-tailed Student's *t*-test (sham vs. 30dd MCAO).

The online version of this article includes the following figure supplement(s) for figure 3:

**Figure supplement 1.** Thbs4 astrocytes derive from slow proliferative cells after brain ischemia.

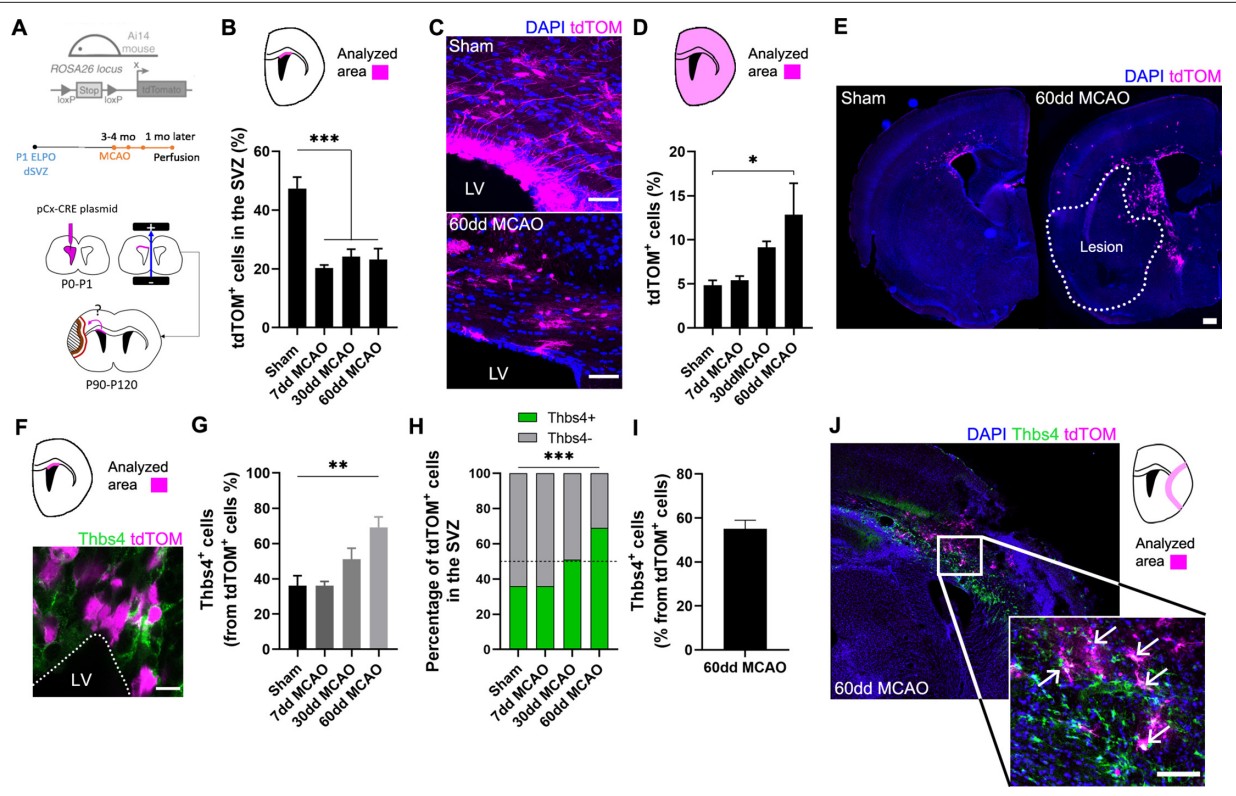

**Figure 4.** Ischemia-induced Thbs4 astrocytes migrate from the subventricular zone (SVZ) to ischemic areas. (**A**) Experimental design: Ai14, ROSA26-CAG-tdTomato transgenic mice were electroporated in the dorsal SVZ at P1. The pCAGGSx-CRE plasmid induced tdTomato (tdTOM) expression in dorsal neural stem cells (NSCs). Middle cerebral artery occlusion (MCAO) was performed 3–4 months later in the same electroporated hemisphere. Mice were analyzed at 7, 30, and 60 dpi. (**B**) Quantification of tdTOM-positive cells shows a decrease in the dorsal SVZ following brain ischemia. (**C**) Representative images of tdTOM-positive cells in sham (top) and 60 dpi SVZ (bottom). (**D**) Quantification of tdTOM-positive cells in the whole brain shows a gradual increase outside the SVZ after brain ischemia. (**E**) Representative images of tdTOM expression in sham (left) and 60 dpi mice (right). (**F**) Representative image of Thbs4/tdTOM-positive cells in the dorsal SVZ. (**G**) Quantification shows an increase in Thbs4/tdTOM-positive cells in the SVZ at 30 and 60 dpi. (**H**) The proportion of Thbs4-positive cells within the tdTOM-positive pool increases in the SVZ at 30 and 60 dpi. (**I**) Around 50% of tdTOM-positive cells expressed Thbs4 in the infarcted tissue at 60 dpi. (**J**) Representative image of Thbs4 and tdTOM expression near the damaged area. Scale bar = 50 μm (**C**), 100 μm (**E, J**), and 10 μm (**F**). n=6 animals per condition. Bars represent mean ± SEM. *p<0.05, **p<0.01, and ***p<0.001 by chi-square test (**H**) and Tukey post hoc test (after one-way ANOVA was significant at p<0.05).

The online version of this article includes the following figure supplement(s) for figure 4:

**Figure supplement 1.** Ischemic-induced Thbs4 astrocytes migrate preferentially to caudal infarcted areas.

the dorsal SVZ. Three to four months after electroporation, we performed the MCAO, and tdTOM fluorescence was analyzed at 7, 30, and 60 dpi. To confirm the efficient labeling of NSCs by the electroporation protocol, we followed the tdTOM fluorescence in the OB (*Figure 4—figure supplement 1A*, see 'Materials and methods'). Only those animals with tdTOM-positive cells in the OB were taken into account for later analysis. After quantification of tdTOM-positive NSCs 7, 30, and 60 dpi, we observed a significant decrease in tdTOM-positive cells in ischemic animals in all SVZ regions analyzed (*Figure 4B and C*, *Figure 4—figure supplement 1B*), suggesting that they either left the area or died. Conversely, there was a general increase of tdTOM-positive cells outside the SVZ (*Figure 4D and E*), especially in the intermediate and caudal axis of the whole brain compared to sham animals (*Figure 4—figure supplement 1C–F*), suggesting that electroporated cells had migrated away from the SVZ.

To identify the proportion of NSCs that differentiated into Thbs4-positive astrocytes following ischemia, we analyzed the co-expression of Thbs4 with tdTOM after MCAO. Despite the general decrease of tdTOM-positive cells in the SVZ, the fraction of Thbs4/tdTOM cells increased at 30 dpi after MCAO (*Figure 4F and G*), reaching a maximum increase at 60 dpi (*Figure 4H*), in particular in rostral and intermediate SVZ (*Figure 4—figure supplement 1G*). Finally, we analyzed the Thbs4-positive

astrocytes recruitment to the ischemic regions. More than half of the total tdTOM-positive cells in the infarcted area expressed the Thbs4 marker at 60 dpi (*Figure 4I and J*). These results suggest that ischemia-induced astrogliogenesis in the SVZ occurs in type B cells from the dorsal region and that these newborn Thbs4-positive astrocytes migrate to the ischemic areas.

## Thbs4-positive astrocytes accumulate at the glial scar after brain ischemia

Besides its role in the neurogenic niches, Nestin is also rapidly and locally expressed by reactive astrocytes and participates in glial scar formation (*Frisén et al., 1995*). Thus, we used Nestin immunofluorescence as a reliable tool to identify the lesion border where the glial scar occurs (*Figure 5A*). Nestin staining of the scar was also compared with more classical methodologies like TTC staining (*Figure 5B*), where viable tissue is stained red and dead tissue is left unstained. The white-pale pink area was then measured as infarct tissue by TTC staining (*Figure 5C*). Unlike the TTC staining, which can only be performed shortly after the lesion, the visualization of the glial scar with Nestin immunofluorescence allowed us to analyze the lesion at later time points. We observed a rapid formation of the glial scar as early as 7 dpi (*Figure 5D*). Analyzing the expression of the two astrocytic markers, Thbs4 and GFAP, as a percentage of the glial scar region of interest (ROI), we observed different dynamics and localization. GFAP expression increased rapidly after 7 days within the glial scar ROI, whereas Thbs4 was upregulated at 15 dpi (*Figure 5E*). This delay suggested that the glial scar harbors two different astrocytic subpopulations: GFAP-only or GFAP/Thbs4. To analyze the spatial location of GFAP and Thbs4 markers within the Nestin border, we measured the fluorescence intensity profile of damaged area at 30 dpi. GFAP was found in both the scar and core areas, whereas Thbs4 fluorescence was found mostly at the borders of the glial scar (*Figure 5F and G*). Analysis and dynamics of Thbs4-positive astrocyte migration were further performed in the corpus callosum (CC), striatum (Str), cortex (Ctx), and OB (*Figure 5—figure supplement 1A*). We observed a linear increase of Thbs4-positive astrocytes from 7 to 60 dpi, both in the Ctx and Str (*Figure 5—figure supplement 1B*). Migration occurred mainly through the corpus callosum; among the total GFAP-positive astrocytes, 36.5% were also positive for Thbs4, suggesting that more than one-third of the astrocytes in the glial scar originated from the SVZ (*Figure 5—figure supplement 1C*). Likewise, the number of DCX-positive cells dramatically decreased 15 dpi in all OB layers (*Figure 5—figure supplement 1D–F*) together with a significant decrease of BrdU-positive cells and DCX/BrdU-positive neuroblasts in the RMS (*Figure 5—figure supplement 1G and H*). This suggests that ischemia induced a change in the neuroblasts' ectopic migratory pathway, depriving the OB layers of SVZ newborn neurons.

## Thbs4-positive astrocytes modulate extracellular matrix homeostasis at the glial scar

To investigate the role of Thbs4-positive astrocytes in the glial scar, we analyzed HA, the main component of the brain ECM, using the hyaluronic acid binding protein (HABP) as a marker. Following ischemic damage, local astrocytes (GFAP+/Thbs4-) react by adopting a fibrotic morphology, producing high-molecular weight HA (HMW-HA) to form the glial scar and isolate the damaged area (*Preston and Sherman, 2011*). We then examined HA deposition in the corpus callosum and the infarcted areas after MCAO. HABP immunofluorescence (*Figure 6A*) and image analysis revealed HA accumulation in the infarcted areas. Skeleton analysis, which quantifies the length of HA chains and correlates with its molecular weight, identified progressively longer HA cable-like structures at the glial scar as the infarct evolved (*Figure 6B*). Fractal dimension analysis, which reflects the interconnectivity of the ECM (*Soria et al., 2020*), further confirmed HA accumulation and increased matrix complexity in infarcted areas, but not in corpus callosum (*Figure 6C*), suggesting that ECM accumulation is different in white and gray matter.

The proximity of Thbs4-positive astrocytes to HA in the ischemic scar (*Figure 6D and E*) led us to investigate if this population contributed to ECM production and, consequently, to scar formation. Since HA is produced exclusively at the cell membrane (*Sherman et al., 2015*; *Zimmermann and Dours-Zimmermann, 2008*), we analyzed HA labeling within 1 µm of the Thbs4-positive cell membrane in high-resolution confocal images (*Figure 6F*). Image analysis showed a significant increase in HABP area at Thbs4-positive astrocyte-cell membranes after MCAO (*Figure 6G*), suggesting a role of Thbs4-positive astrocytes in HA production within the glial scar.



**Figure 5.** Ischemia-induced Thbs4 astrocytes accumulate at the glial scar. (**A**) The glial scar could be observed at 30 dpi using Nestin immunofluorescence. (**B**) Classical TTC technique underestimated ischemic regions compared with Nestin immunofluorescence. (**C**) Representative images of TTC staining in infarcted brains. (**D**) Representative confocal images showing the time course of Nestin expression at 3, 7, 15, and 30 dpi. (**E**) The Nestin-positive glial scar was established by 15 dpi, while Thbs4 astrocytes arrived at the glial scar later, by 30 dpi. (**F**) Fluorescence profile of Thbs4, GFAP, and Nestin marker from the core to penumbra areas showed Thbs4 was mainly located at the borders of the Nestin-positive glial scar. (**G**) Representative images of Thbs4 (green), GFAP (white), and Nestin (red) markers in 30 dpi mice. Scale bar = 50 μm (**D**, **G**). n=3 animals per condition. Bars/lines represent mean ± SEM. *p<0.05 and **p<0.01 by two-tailed Student's *t*-test (TTC vs. Nestin) and by Dunnett post hoc test (after two-way ANOVA was significant at p<0.05).

The online version of this article includes the following figure supplement(s) for figure 5:

**Figure supplement 1.** Subventricular zone (SVZ) neural stem cells (NSCs) stop producing newborn neurons to send newborn astrocytes to the ischemic areas.

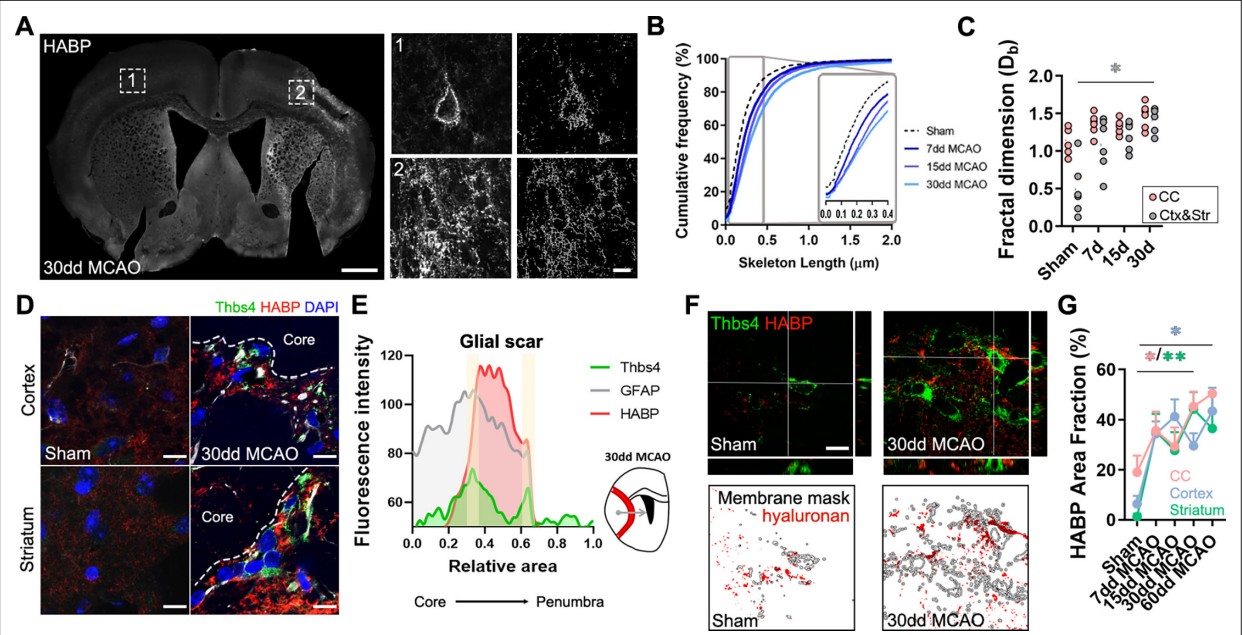

**Figure 6.** Thbs4-positive astrocytes produce hyaluronan at the glial scar after middle cerebral artery occlusion (MCAO). (**A**) Representative images of hyaluronic acid (HA) labeled with HABP in the contralateral (1) and ipsilateral hemisphere (2). Only perineuronal nets are clearly visible in the healthy tissue, while a dense interstitial matrix is observed in ischemic areas. (**B**) Cumulative distribution of skeletonized HA signal in sham and infarcted regions of 7, 15, and 30 dpi mice (p<0.0001). (**C**) Quantification of fractal dimension showed increased extracellular matrix interconnectivity in the infarcted cortex and striatum (Ctx and Str) at 30 dpi. (**D**) Representative confocal images of Thbs4 (green) and HABP (red) in the cortex and striatum of sham and 30 dpi mice. Note the HA labeling on the surface of Thbs4-positive cells. (**E**) Fluorescence intensity profile of Thbs4, GFAP, and HABP markers from the lesion core to penumbra areas. Thbs4 expression was mainly restricted to the borders of the HA-positive glial scar at 30 dpi. (**F**) Representative confocal images of Thbs4 and HABP markers in sham and 30 dpi mice (top). Examples of HA (red) occupying the membrane mask of Thbs4-positive astrocytes in sham and 30 dpi mice (bottom), showing increased HA synthesis after MCAO. (**G**) Quantification of membrane-bound HA showed increased HA along Thbs4-positive membrane after brain ischemia (corpus callosum: pink; infarcted cortex: blue; infarcted striatum: green). Scale bar = 500 µm (**A**), 10 µm (inset in **A**, **D**, and **F**). n=6 animals per condition. *p<0.05 and ****p<0.0001 by Kruskal–Wallis test and two-way ANOVA (significant at p<0.05).

The online version of this article includes the following figure supplement(s) for figure 6:

**Figure supplement 1.** Thbs4 astrocytes internalize hyaluronic acid (HA) in the infarcted areas.

**Figure supplement 2.** In vitro neural stem cell (NSC)-derived Thbs4 cells synthesize hyaluronic acid (HA) after a hypoxic-ischemic condition.

To explore the potential role of Thbs4-positive astrocytes in HA removal, we quantified intracellular HABP spots in Thbs4-positive astrocytes at the glial scar. While HMW-HA cleavage occurs extracellularly, HA fragments can be internalized for degradation by lysosomal hyaluronidases (*Preston and Sherman, 2011*; *Zimmermann and Dours-Zimmermann, 2008*). Intracellular HABP spots were compared between Thbs4-positive astrocytes and local astrocytes (GFAP-positive/Thbs4-negative) (*Figure 6—figure supplement 1A*). Thirty days after MCAO, Thbs4-positive astrocytes showed more intracellular HABP than local astrocytes in the infarcted cortex (*Figure 6—figure supplement 1B*). However, the number of HA spots internalized was higher in local astrocytes compared with Thbs4 astrocytes (*Figure 6—figure supplement 1C*), suggesting that Thbs4-positive cells internalize HA in fewer vesicles. To confirm that astrocytes were degrading HA, we performed an immunofluorescence analysis of CD44 (*Figure 6—figure supplement 1D*). CD44 is a transmembrane glycoprotein receptor that mediates HA internalization and degradation, and therefore ECM remodeling (*Dzwonek and Wilczynski, 2015*). We detected that around 50% of the total Thbs4-positive astrocytes contained internalized CD44 receptor in the striatum 30 dpi (*Figure 6—figure supplement 1E*), with higher numbers (60%) for the infarcted corpus callosum. These findings suggest that Thbs4-positive astrocytes participate in both the formation and degradation of the ECM at the glial scar.

To further investigate HA internalization by astrocytes, we conducted in vitro experiments using cultures of rat SVZ neural stem cells, a more versatile preparation that displays the same HA internalization and degradation machinery as mouse brain tissue (*Sherman et al., 2015*). An oxygen and

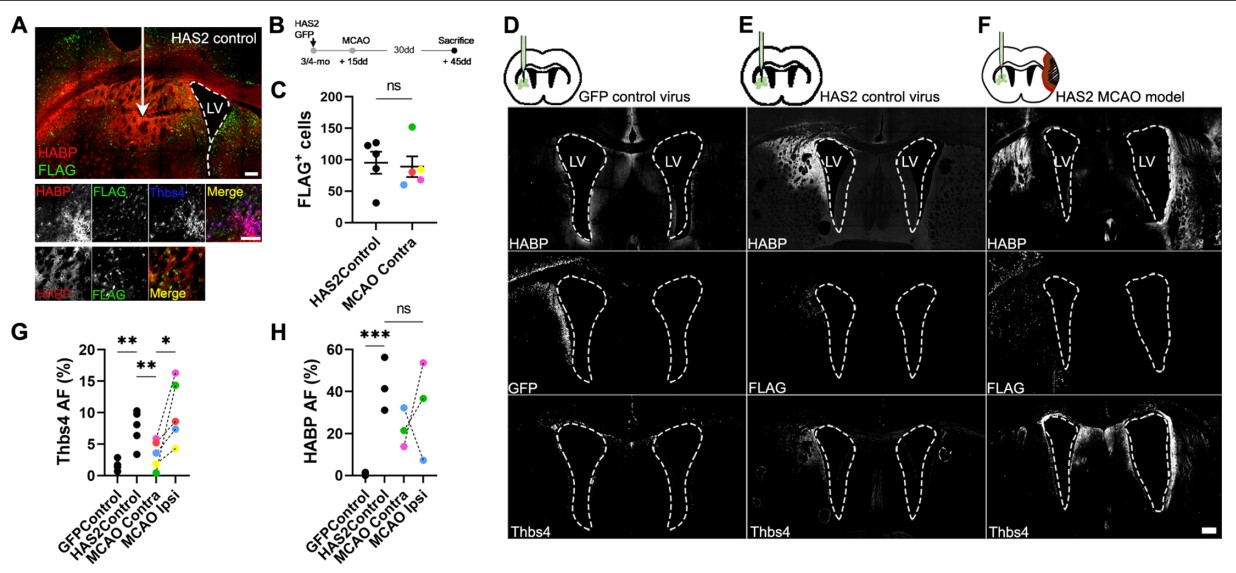

**Figure 7.** Hyaluronan accumulation is sufficient but not exclusive to recruit Thbs4 astrocytes. (**A**) Representative images of hyaluronic acid (HA) accumulation caused by viral-mediated overexpression of Has2 from naked mole rat. (**B**) Experimental design: Has2 viral injections were performed 2 weeks before middle cerebral artery occlusion (MCAO). Animals were sacrificed 30 days after MCAO. We used standard GFP virus as control. Two groups of Has2-injected mice were studied, with only one subjected to MCAO. (**C**) FLAG-positive cells (Has2 virus tag) did not show changes in either Has2 control (Has2Control) or Has2-MCAO group (contralateral hemisphere). (**D–F**) Representative images of HABP, GFP, and Thbs4 markers in GFP control virus (**D**), HAS2 control virus (**E**), and Has2 MCAO group (**F**). (**G**) Thbs4 expression increased after Has2 viral-mediated overexpression. However, Thbs4 was upregulated in the ischemic hemisphere when MCAO was performed together with the Has2 viral infection (MCAO ipsi). GFP control virus did not induce an increase of Thbs4. (**H**) HABP was used as a control for HA accumulation. Thbs4 astrocytes in Has2 MCAO group increased in the ischemic hemisphere even though HA accumulation in both hemispheres (HAS2 and MCAO hemisphere), suggesting additional ischemia-induced signals recruit Thbs4 astrocytes to infarcted areas. Scale bar = 100 µm (**A, F**). n=6 animals per condition. *p<0.05, **p<0.01, and ***p<0.001 by two-tailed Student's *t*-test (Has2Control vs. MCAO Contra) and Tukey post hoc test (after one-way ANOVA was significant at p<0.05).

glucose deprivation (OGD) protocol was applied as an ischemia model and cells were fixed 7 days after OGD (*Figure 6—figure supplement 2A*). HA was analyzed via HABP immunofluorescence, both inside and outside the Thbs4-positive cells (*Figure 6—figure supplement 2B and C*). Thbs4-positive cells showed increased intracellular HA after OGD, while extracellular HA levels decreased (*Figure 6— figure supplement 2D and E*). However, this phenomenon was reversed after adding hyaluronidases in the extracellular space (*Figure 6—figure supplement 2F and G*), suggesting that Thbs4-positive cells from SVZ homogenates can internalize or synthesize HA depending on external stimuli. HA internalization was also measured in Iba1-positive cell cultures where we observed a reduced internalization compared to Thbs4-positive cells (*Figure 6—figure supplement 2H*). Lastly, metalloproteinase activity was found to be increased in Thbs4 cultures following OGD (*Figure 6—figure supplement 2I*). Altogether, these results suggest Thbs4-positive cells can internalize HA after ischemic conditions but also synthesize it when ECM is degraded.

## Hyaluronan accumulation is sufficient but not exclusive for the recruitment of Thbs4-positive astrocytes

Since Thbs4-positive astrocytes migrate to the glial scar but not to the ischemic core, we hypothesized that HA, the primary component of the ischemic scar, may be the trigger signal for recruiting Thbs4-positive astrocytes. To test this hypothesis, we simulated the ECM accumulation that occurs at the scar by overexpressing the Hyaluronan synthase 2 (Has2) from the naked mole rat (*Heterocephalus glaber*), a tool often used in mice to produce HA of very high molecular weight (*Tian et al., 2013*; *Zhang et al., 2023*). We used a viral vector (AAV2/9-CAG-Has2-3xFlag) with a FLAG tag to visualize transduced cells after intracerebral inoculation in the striatum and checked HA accumulation in FLAG-positive areas using HABP staining (*Figure 7A*). We overexpressed Has2 (or GFP as control) unilaterally. In a subset of animals that underwent MCAO, AAV-Has2 was injected contralateral to the infarct, to compare Thbs4-positive astrocytes recruitment by both stimuli, the inflammation from the

ischemic damage and the HA accumulation per se (*Figure 7B*). FLAG-positive cells were quantified as a surrogate marker of viral infection for both control and MCAO groups (*Figure 7C*). The GFP control group did not increase HA production per se (*Figure 7D*). However, HA overexpression was observed in Has2 groups (both control and MCAO group) by HABP immunofluorescence (*Figure 7E and F*).

HA accumulation was sufficient to recruit the Thbs4-positive astrocytes from the ipsilateral SVZ 45 days after AAV-Has2 inoculation (*Figure 7G*), with a similar time window to that observed after MCAO. Interestingly, when we induced a lesion by MCAO and performed viral overexpression of Has2 in the contralateral hemisphere, we found that Thbs4-positive astrocytes preferentially migrated to the ischemic glial scar (*Figure 7G*). This migration of Thbs4-positive astrocytes to the site of ischemic damage occurred despite no differences in HA overexpression between contralateral (Has2 infected) and ipsilateral (infarcted area) hemispheres (*Figure 7H*). These results suggest that the accumulation of HMW-HA alone is sufficient to activate and recruit Thbs4-positive astrocytes from the SVZ. However, the preferential migration to the ischemic scar indicates that brain ischemia generates a more potent biochemical signal that not only activates the SVZ but also recruits newborn astrocytes.

## Ischemia-induced gliogenesis at the SVZ correlates with local hyaluronan degradation

It is known that the niche microenvironment has a unique ECM composition (*Kjell et al., 2020*) and that the stiffness of the niche plays a crucial role in the proliferation and differentiation of progenitor cells (*Long and Huttner, 2019*; *Segel et al., 2019*). After observing the close association between Thbs4-positive astrocytes and the ECM at the glial scar, we decided to study the Thbs4/HA tandem in the SVZ following MCAO (*Figure 8A*). We did not observe any significant changes in the HA area across the entire SVZ but noted a decrease at 7 and 15 dpi in the dorsal SVZ (*Figure 8B–D*). Skeleton analysis showed fragmentation of HA exclusively in the dorsal SVZ at 7 and 15 dpi (*Figure 8E–G*), that is, earlier than the migration of these newborn astrocytes toward the lesion.

Next, we examined the expression of genes related to HA degradation (*Hyal1*, *Hyal2,* and *Hyal3*) and synthesis (*Has1* and *Has2*). We also analyzed *Thbs4* and hypoxia-inducible factor 1-alpha (*Hif1a*), which are known to be upregulated in the SVZ following brain ischemia (*Benito-Muñoz et al., 2016*; *Liu et al., 2007*). Fresh SVZ tissue was collected from sham, 15 and 30 dpi mice (*Figure 8H*). We observed an increased expression of all hyaluronidases (*Hyal*) at 15 dpi and slightly decreased by 30 dpi, whereas synthases (*Has*) showed a gradual increase from 15 to 30 dpi (*Figure 8I*). Additionally, we observed an upregulation of *Thbs4* and *Hif1a* both at 15 and 30 dpi (*Figure 8I*). These results suggest that HA degradation in the SVZ is correlated with the Thbs4 response to ischemia.

## Discussion

The effects of brain insult on adult neurogenesis have been extensively discussed, both in animal models and humans. However, whether these injuries can trigger the generation of newborn astrocytes from the neurogenic niches remains a topic of debate. Neural stem cells in the adult brain are a remnant of embryonic radial glia. Some researchers suggest that depletion of neurogenesis in the adult brain is a consequence of astrogliogenesis (*Encinas et al., 2011*), while others propose astrogliogenesis is part of a neurorestorative program originating from the neurogenic niches (*Bonzano et al., 2018*; *Sohn et al., 2015*). Despite these differing perspectives, direct and definitive evidence of the generation of newborn astrocytes from the SVZ or the SGZ has yet to be presented, as well as the potential role of these astrocytes in aging and pathological conditions. In this study, we confirm that a strong and acute damage-associated stimulus in the brain, such as an ischemic infarct, prompts astrogliogenesis in the SVZ. Furthermore, we demonstrate that these astrocytes, which are positive for Thbs4, migrate to the glial scar and play a role in its modulation, being capable of synthesis and degradation of hyaluronan, a key component of the ECM at the scar.

Our findings align with previous studies, where cortical injury initiated by photothrombotic/ischemia was shown to trigger a substantial production of Thbs4-positive astrocytes from the SVZ, a process modulated by Notch activation (*Benner et al., 2013*). In this work, the authors suggested a clear role for Thbs4-positive astrocytes in modulating the ischemic scar. In fact, homozygotic deletion of the Thbs4 gene led to abnormal glial scar formation and increased microvascular hemorrhage following brain ischemia. Similarly, our results demonstrate that the adult neurogenic niche at the SVZ



**Figure 8.** Hyaluronic acid (HA) degradation in the subventricular zone (SVZ) correlates with Thbs4 response to brain ischemia. (**A**) Representative image of Thbs4 and HA (HABP) markers in the SVZ at 30 dpi. (**B**) Representative images of Thbs4 and HA in the sham and 15, 30 dpi SVZ. (**C, D**) HA levels did not change in the entire SVZ after middle cerebral artery occlusion (MCAO) (**C**) but decreased significantly in the dorsal SVZ (**D**). Thbs4 expression increased throughout the SVZ after MCAO (**C**), with a marked rise at 30 dpi in the dorsal SVZ (**D**), coinciding with HABP reduction. (**E**) Representative confocal (top) and skeletonized (bottom) images of HABP in the dorsal SVZ of sham, 15 and 30 dpi mice. (**F**) Cumulative distribution of HA skeletons showed a reduction in the length of HA cable-like structures only in the dorsal SVZ at 7 and 15 dpi. (**G**) The interconnectivity of the extracellular matrix decreased by 7 dpi in the dorsal SVZ, as measured by fractal dimension. (**H**) Experimental design for qPCR: 3–4-month-old mice were subjected to

*Figure 8 continued on next page*

*Figure 8 continued*

MCAO, and fresh SVZ tissue was extracted 15 and 30 days after MCAO for RT-qPCR analysis. (**I**) Hyaluronan degradation genes (*Hyal1, Hyal2,* and *Hyal3*) increased by 15 and 30 dpi in the SVZ, while hyaluronan synthase genes (*Has1* and *Has2*) were overexpressed later, at 30 dpi. *Thbs4* and *Hif1a* were also upregulated by 15 and 30 dpi in the SVZ. Scale bar = 100 µm (**A**), 20 µm (**B, E**). n=6 (**C–E** and **G**) and 3 (**I**). *p<0.05, **p<0.01, and ***p<0.001 by Kruskal–Wallis test and Tukey post hoc test (after one-way ANOVA was significant at p<0.05).

responds early after ischemic stroke, degrading local hyaluronan plausibly to modify niche stiffness and facilitate proliferation and differentiation of NSCs. Further experiments using Thbs4-knockout models will help confirm this hypothesis.

By labeling NSC via in vivo postnatal electroporation, we provided proof of the origin and migration of newborn Thbs4-positive astrocytes from the SVZ to the ischemic scar in response to the injury. Notably, Thbs4 is expressed in the newborn astrocytes that migrate to the scar, whereas local reactive astrocytes at the site of injury express only GFAP. It is likely that SVZ-derived newborn astrocytes require Thbs4 expression to facilitate their migration to the ischemic area, as Thbs4 deletion has been shown to impair migration of SVZ-derived cells along the RMS (*Girard et al., 2014*).

It is noteworthy that while the local, 'GFAP-only', reactive astrocytes at the scar are capable of forming the required ECM (*Cregg et al., 2014*; *Wanner et al., 2013*), we show that Thbs4-positive (Thbs4/GFAP) astrocytes are also recruited from the SVZ to modulate scar formation. Unlike local reactive astrocytes, which are hastily generated and present both at the core and periphery of the lesion, the newborn astrocytes from the SVZ arrive later to the injury. We demonstrate that these Thbs4-positive astrocytes migrate from the SVZ to the lesion and halt at the lesion border (precisely defined by Nestin staining), establishing residence at the scar but not beyond.

The molecular signals that trigger gliogenesis in the SVZ and recruit newborn astrocytes to the lesion remain unclear. We previously identified adenosine, a by-product of ATP hydrolysis, as a strong candidate, as it stimulates astrogliogenesis through A1 receptors following OGD in vitro and ischemia in vivo (*Benito-Muñoz et al., 2016*). Here, we show that viral-mediated overproduction of HA, the primary component of the glial scar, is sufficient to activate the generation and migration of the Thbs4-positive astrocytes from the SVZ. Loss-of-function studies, using HA-depletion models or HA synthase (Has)-deficient mice, are still needed to corroborate this finding, although the inflammation associated with HA deficiency might confound interpretation. Indeed, our results also show that the ischemic infarct, with its proinflammatory (cytokines, chemokines) and proteolytic (MMPs, ROS) signals (*Jayaraj et al., 2019*), serves as a more potent attractor than the ECM alone.

The glial response from the SVZ we propose in this study might have several functional implications. Thbs4-positive astrocytes generated from the SVZ might act as a second wave of reactivity, supporting local astrocytes in modulating ECM formation. Hyper-reactive astrocytes at the lesion border decrease over time, along with reductions in HA synthesis and CD44 activity, suggesting that the second wave of Thbs4-positive astrocytes from the SVZ could sustain high levels of HA production (*Lindwall et al., 2013*). Thus, Thbs4-positive astrocytes may be crucial during this delayed temporal window when glial scar remodeling is essential for neurological recovery (*Cai et al., 2017*; *Dzyubenko et al., 2018*). However, it's important to note that only about one-third of the astrocytes at the glial scar are derived from the SVZ. This suggests that while SVZ-derived astrocytes may be important for glial scar maintenance, local astrocytes remain the central players in this process.

Based on our observations of both HA production and internalization by Thbs4-positive astrocytes in vivo within the ischemic region and in vitro following OGD, we propose a role for these astrocytes in the modulation, and possibly maintenance, of the ECM at the scar. It is likely that ECM degradation at the scar is also carried out by professional macrophages, such as microglia, which are known to be recruited to the scar (*Fawcett and Asher, 1999*) and have been shown to degrade ECM through phagocytosis (*Nguyen et al., 2020*; *Soria et al., 2020*). Understanding which cells within the scar environment participate in its modulation, particularly its degradation, holds highly therapeutic value for promoting regeneration (*Dzyubenko et al., 2018*; *Silver and Miller, 2004*). Here, we shed light on the unknown role of SVZ-derived astrocytes activated by ischemic brain injury and propose them as critical players in tissue regeneration after brain ischemia.

## Materials and methods

### Animals

Male and female wild-type C57BL/6J mice and Ai14-Rosa26-CAG-tdTomato^flx/flx transgenic mice bred at the animal facility of UPV/EHU were used for in vivo experiments. Sprague–Dawley rat pups from P4 to P7 were used for in vitro experiments. Animals were maintained under standard laboratory conditions with food and water ad libitum. All animals were handled in accordance with the European Communities Council Directive (2010/63/EU) and approved by the Ethics Committee of the University of the Basque Country (UPV/EHU) under license CEEA/340/2013.

### Transient MCAO

Brain ischemia was induced in 3–4-month-old C57BL/6 mice (approximately 25 g) by occluding 60 minutes of the lenticulo-striate arteries with a 10 mm length of silicone-coated 6-0 monofilament nylon suture (602223PK10, Doccol Corporation) using a modification of the protocol described by *Gelderblom et al., 2009*. Artery occlusion generated a specific lesion in the cortex and striatum that was evaluated at different times after the MCAO (7, 15, 30, or 60 days after occlusion). Ischemic damage was evaluated in the ipsilateral hemisphere by Cresyl violet staining, Nestin immunofluorescence, or hyaluronan staining. Sham animals, in which arteries were visualized but not disturbed, were used as an experimental control. Both experimental groups, sham and ischemic, were anesthetized with a mix of isoflurane and oxygen (induction of anesthesia with 4% isoflurane and maintenance for surgery at 1.5% isoflurane) and maintained under analgesia with buprenorphine 24 hours after ischemia (0.03 mg/kg body weight i.p. every 12 hours). Animal wellness was evaluated daily, controlling the body weight and the motor deficit of the animal by means of neurological scores and pole test. Motor deficit was assessed in each animal 1 hours after MCAO and later every 24 hours. Animals without neurological deficits were excluded from the study.

### 5-bromo-2'-deoxyuridine (BrdU) protocol

The total pool of proliferating cells was labeled with three injections of 50 mg/kg 5-bromo-2'-deoxyuridine (BrdU) the day before MCAO protocol. Animals were sacrificed the day after MCAO to analyze acute SVZ proliferation. Chronic BrdU protocol was performed for quiescent NSC identification (*Codega et al., 2014*). Briefly, 1% of BrdU was administered in the drinking water for 14 days. On day 15, BrdU was removed and animals were subjected to the MCAO protocol. Mice were sacrificed and perfused at 3, 7, 15, and 30 days after MCAO, although only animals with a temporal window of 30 days between MCAO and perfusion were used for statistical analysis. Activated cells were labeled by i.p. injection of 50 mg/kg 5-iodo-2'-deoxyuridine (IdU) 24 hours before euthanasia.

### Electroporation

Ai14-Rosa26-CAG-tdTomato^flx/flx transgenic mice and pCAGGS-CRE plasmid were kindly provided for Prof. Harold Cremer and Marie-Catherine Tiveron from the Institut de Biologie du Développement de Marseille (IBDM). Postnatal electroporation was performed as described previously (*Platel et al., 2019*). Mice pups from P0-P2 were anesthetized by hypothermia. pCAGGS-CRE plasmid was used at a final concentration of 2.5 μg/ul and 2 μl was injected in the lateral ventricle by expiratory pressure using a glass micropipette. Following injection, pup brains were placed between electrodes targeted at the dorsal SVZ and subjected to five 95 V electrical pulses (50 ms, 950 ms intervals). BTX ECM399 (Thermo Fisher Scientific) electroporator was used with 7 mm tweezer electrodes. Electroporated pups were reanimated at 37°C before returning to the mother.

In order to analyze the Thbs4-positive cells in the SVZ, we subcloned the eGFP gene (from a pCMV-eGFP N1) into a PGL3-basic backbone containing mouse Thbs4 promoter, kindly provided by Stenina-Adognravi lab (*Muppala et al., 2017*). pThbs4-GFP was also electroporated in P0-P2 mice and euthanized 1 day post-electroporation to corroborate Thbs4 expression in the SVZ at postnatal stages.

### AAV vector production

AAV2/9-CAG-Has2-3xFlag vector was produced by polyethylenimine (PEI)-mediated triple transfection of low-passage HEK-293 T/17 cells (ATCC; Cat# CRL-11268). The respective AAV expression

plasmid expression plasmid pNMRHas2_N1 (provided by Dr. Vera Gorbunova) was cotransfected with the adeno helper pAd Delta F6 plasmid (Penn Vector Core, Cat# PL-F-PVADF6) and AAV Rep Cap pAAV2/9 plasmid (Penn Vector Core, Cat# PL-TPV008). AAV vector was purified as previously described (*Bourdenx et al., 2015*). Cells are harvested 72 hours post-transfection, resuspended in lysis buffer (150 mM NaCl, 50 mM Tris–HCl pH 8.5), and lysed by three freeze–thaw cycles (37°C/–80°C). The cell lysate is treated with 150 units/ml Benzonase (Sigma, St Louis, MO) for 1 hours at 37°C, and the crude lysate is clarified by centrifugation. Vectors are purified by iodixanol step gradient centrifugation and concentrated and buffer-exchanged into Lactated Ringer's solution (Baxter, Deerfield, IL) using vivaspin20 100 kDa cut-off concentrator (Sartorius Stedim, Goettingen, Germany). Titrations were performed at the transcriptome core facility (Neurocentre Magendie, INSERM U862, Bordeaux, France). The genome-containing particle (gcp) titer was determined at a concentration of $1.10^{13}$ gcp/ml by quantitative real-time PCR using the Light Cycler 480 SYBR green master mix (Roche, Cat# 04887352001) with primers specific for the AAV2 ITRs (fwd 50-GGAACCCCTAGTGATG GAGTT-3'; rev 50-CGGCCTCAGTGAGCGA-30) on a Light Cycler 480 instrument. The purity assessment of vector stocks was estimated by loading 10 µl of vector stock on 10% SDS acrylamide gels; total proteins were visualized using the Krypton Infrared Protein Stain according to the manufacturer's instructions (Life Technologies).

## In vivo AAV-HAS2 injections

Mice were anesthetized with isoflurane and fixed to a stereotaxic frame connected with a gas mask. We injected 1 µl of adeno-associated virus (AAV2/9) CAG-Has2-3xFlag ($1 \times 10^{13}$ gc/µl) in the contralateral striatum (AP:+1.0; ML: –1.8; DV: –3.2) to simulate scar tissue hyaluronan. After 2 weeks, the ischemic group was subjected to the MCAO model in the ipsilateral hemisphere and perfused 30 days later. Only slices with representative lesions in both hemispheres were used for image and statistical analysis.

**Table 1.** Primary antibodies used for in vivo immunofluorescence protocol.

| Antibody | Host | Isotope | Dilution | Reference | Company |
|---|---|---|---|---|---|
| Thbs4 | Mouse | Monoclonal IgM kappa light chain | 1/400 | Sc-390734 | Santa Cruz Biotechnology, USA. |
| Thbs4 | Goat | Polyclonal IgG | 1/200 | AF2390 | R&B Systems, USA |
| S100β | Rabbit | Polyclonal | 1/400 | Z0311 | Dako, Agilent Technologies, Inc |
| S100β | Mouse | Monoclonal IgG2a | 1/1000 | MA5-12969 | Invitrogen, Spain |
| GFAP | Mouse | Monoclonal IgG1 | 1/1000 | MAB3402 | Sigma, Spain |
| GFAP | Rabbit | Polyclonal | 1/4000 | Z0334 | Dako, Agilent Technologies, Inc |
| βIII tubulin | Rabbit | Polyclonal | 1/400 | ab18207 | Abcam, UK |
| NeuN | Mouse | Monoclonal IgG1 | 1/1000 | MAB377 | Millipore, Germany |
| DCX | Rabbit | Polyclonal IgG | 1/500 | ab18723 | Abcam, UK |
| Nestin | Chicken | Polyclonal IgY | 1/1000 | NES | Aves Lab, EEUU |
| Prominin 1 | Rat | Monoclonal IgG1 kappa | 1/300 | 14-1331-82 | Invitrogen, Spain |
| EGFR | Mouse | Monoclonal | 1/400 | SAB4200809 | Abcam, UK |
| BrdU | Rat | Monoclonal | 1/400 | MCA2060 | Bio-Rad, USA |
| BrdU | Mouse | Monoclonal IgG1, kappa | 1/500 | 03-3900 | Invitrogen, Spain |
| IdU | Mouse | Monoclonal IgG2b | 1/100 | MA5-24879 | Invitrogen, Spain |
| Ki67 | Rabbit | Polyclonal IgG | 1/400 | Ab15580 | Abcam, UK |
| Cleaved-Caspase 3 | Rabbit | Monoclonal IgG | 1/300 | #9664 | Cell Signaling, Germany |
| CD44 | Rat | Monoclonal IgG2b | 1/500 | MA517875 | Invitrogen, Spain |
| FLAG | Mouse | Monoclonal IgG1 | 1/500 | F1804 | Sigma, Spain |
| Olig2 | Mouse | Monoclonal IgG2aк | 1/1000 | MABN50 | Millipore, Germany |

## Immunofluorescence

Mice were anesthetized with an intraperitoneal injection of pentobarbital (100 mg/kg). Brain tissues were fixed by intracardiac perfusion using 4% paraformaldehyde. Fixed brains were sliced at 40 μm of thickness using the Leica VT1200S vibratome (Leica Microsystems) and subjected to immunofluorescence. For standard immunofluorescence protocol, tissue slices were permeabilized with 0.1% Triton and non-specific epitopes blocked with 1% bovine serum albumin (BSA) and 10% donkey serum in PBS. Primary antibodies were incubated at different concentrations (see *Table 1* for specification) overnight at 4°C and then washed three times with 0.1% Triton in PBS. Secondary conjugated antibodies were incubated for 1 hour at room temperature. After three washes with 0.1% Triton in PBS, cells were stained for 10 minutes with DAPI and further washed with PBS. Finally, coverslips were mounted with Fluoromount (SouthernBiotech) and analyzed by fluorescence using the confocal microscope Leica TCS SP8. For HA labeling, endogenous biotins were pre-blocked using Streptavidin/Biotin blocking solution (Vector Labs, #SP-2002) following the manufacturer's instructions and further blocked with PBS 1X0.1% Saponin 1% BSA for 1 hour at room temperature. Primary antibodies and biotinylated Hyaluronic Acid Binding Protein (HABP, #385911 Millipore) were incubated for 72 hours at 4°C or 24 hours at room temperature. After three washes with PBS, 1X0.1% Saponin, secondary antibody, and DAPI diluted in the same blocking buffer were incubated for 1 hour at room temperature. Slices were mounted with Mowiol (Millipore #475904) in superfrost slides (Thermo Fisher Scientific) and #1.5 coverslips.

## Imaging and image analysis

Confocal images were acquired in a Leica TCS SP8 with ×20 (0.8 NA) and ×63 objectives (1.4 NA). Images were taken with a pixel size of 90 nm. Widefield imaging was performed in a Zeiss AxioVision fluorescence microscope and a 3DHISTECH panoramic MIDI II digital slide scanner. Settings were adjusted for each individual experiment and kept for all conditions. Image processing was performed using FIJI (*Schindelin et al., 2012*), CaseViewer, and Hyugens (SVI, The Netherlands) software. For Thbs4, Nestin, and DCX immunofluorescence, wholemount SVZ preparations and AAV-HAS2 analysis, tile-scan was done using the navigator plug-in in the LasX software. For Nestin immunofluorescence, images were deconvoluted using the Hyugens software.

SVZ images were reconstructed using the tile-scan tool (LasX software, Leica). SVZ ROI was delimited manually and expression of each marker was measured in the SVZ ROI. Quantification of all markers was done manually, except for HABP quantification, which was done with custom FIJI scripts for in vitro (https://github.com/MariArdaya/Hyaluronan_v2, copy archived at *Ardaya, 2025*) and in vivo analysis (https://github.com/SoriaFN/Tools, copy archived at *Soria, 2025*). The HABP signal was segmented with an automatic threshold and colocalized with other markers such as CD44. This, combined with cell segmentation, based on the Thbs4 mask, allowed for quantification of HABP area and number of spots inside cells, outside cells, or at the membrane (with a band ROI of 1 μm). The Analyze Skeleton and Fractal count plugins were used on the skeletonized segmented HABP to quantify the longest shortest path and the fractal dimension, respectively, to reveal the length and interconnectivity of the matrix network (*Casey et al., 2024*; *Soria et al., 2020*).

## Western blot

The SVZ was freshly dissected on ice and minced using a micro grinder. Total protein was extracted after tissue lysis with RIPA buffer (Thermo Fisher Scientific #89900) and Protease and Phosphatase Inhibitor Cocktail 100X (Thermo Fisher Scientific). Samples were lysed in ice for 10 minutes and sonicated using an Ultrasonic Processor (Hielscher Ultrasound Technology). Sonicated tissues were centrifuged at 1200 rpm for 10 minutes and 4°C to remove insoluble fragments. Protein concentration was measured by chemiluminescence using RC DC Protein Assay (Bio-Rad). Then, 40 μg of protein samples were denatured in reducing loading buffer containing β-mercaptoethanol (M3148, Sigma) and 0.0002% bromophenol blue at 95°C for 5 minutes and separated in pre-casted 12% or 7.5% Tris-Glycine polyacrylamide gel using the Criterion cell system (Bio-Rad). Electrophoresis was conducted at 80 V in a Tris-Glycine buffer (25 mM Tris, 192 mM glycine, 0.1% SDS in $dH_2O$, pH 8.3). After complete separation, proteins were transferred to a nitrocellulose membrane (Bio-Rad) using Trans-Blot Turbo Transfer System (Bio-Rad). Nitrocellulose membranes were incubated in blocking solution (Tris-buffer solution, 0.1% Tween-20 and 5% BSA) for 1 hour at room temperature. Primary antibodies

**Table 2.** Primer sequences for qPCR.

| Name | Sequence |
| --- | --- |
| Hyal1_FW | GTGCCAAGCCCTATGCTAATAAG |
| Hyal1_REV | GCATGTCCATTGCAAAGACTGA |
| Hyal2_FW | GTCCCACATACACCCGAGGA |
| Hyal2_REV | GGCACTCTCACCGATGGTAGA |
| Hyal3_Ffw | GGACGACCTGATGCAGACTATTG |
| Hyal3_REV | GGTCCCCCCAGAGTACCACT |
| Has1_FW | AGGGCTCTTAAAGGAGGAGTCC |
| Has1_REV | AGAAGGTAAACTGAGTCCCCAGAA |
| Has2_FW | CAAAGAGGTTCGTTCAAGTTCTGA |
| Has2_REV | TGTGTTTGTTTCCCACTAGCTCTC |
| Hif1a_FW | CATAAAGTCTGCAACATGGAAGGT |
| Hif1a_REV | ATTTGATGGGTGAGGAATGGGTT |
| HPRT1_FW | TGATAGATCCATTCCTATGACTGTAGA |
| HPRT1_REV | AAGACATTCTTTCCAGTTAAAGTTGAG |

were incubated overnight at 4°C. Fluorescent or horseradish peroxidase-conjugated secondary antibodies were incubated in blocking buffer for 1 hour at room temperature. Immunodetection was performed using chemiluminescence or fluorescence on a Chemidoc MP (Bio-Rad) and the intensity of the bands was quantified using ImageLab (Bio-Rad) software corrected by Gaussian curves.

## qRT-PCR

Fresh SVZ tissue from ipsilateral hemisphere was isolated as previously described from sham, 15 and 30 days after MCAO animals to analyze the expression of several genes (*Table 2*). After mechanical trituration, RNA was extracted from the tissue following the manufacturer's protocol (NZYTECH Total RNA Isolation kit protocol, NZYTECH, MB13402). RNA concentration was measured by spectrophotometry using the Nanodrop 2000c spectrophotometer (Thermo Scientific, USA). Real-time polymerase chain reaction (RT-PCR) from 10 ng of RNA was performed using the One-step NZYTECH RT-qPCR Green kit according to the manufacturer's protocol (NZYTECH, MB34302). RT-PCR was normalized using the housekeeping genes hypoxanthine phosphoribosyl-transferase 1 (HPRT1, QIAGEN Quanti-Tect Primer Assay, Barcelona, Spain). Specificity of each amplification was confirmed by melting curve analysis. PCR reactions were performed in a CFX96 Detection System (Bio-Rad, Madrid, Spain). Semi-quantification was performed using the 2-ΔΔCt algorithm. Unsupervised hierarchical distance matrix and distance matrix map was developed using Orange3 software (Python v3.0).

## Neural stem cell culture

Neurosphere cultures were prepared from 4 to 7-day-old Sprague–Dawley rat pups as previously described (*Cavaliere et al., 2012*). Briefly, the SVZ was isolated and minced with a McIllwain tissue chopper (Campden Instruments). SVZs from 2 to 3 brains were digested for 7 minutes at 37°C in 5 ml of trypsin/EDTA (Sigma). Digestion was stopped by adding an equal volume of trypsin inhibitor (Sigma) and 0.01% DNAse I (Sigma). The cell suspension was centrifuged and the pellet mechanically dissociated in NeuroCult medium (StemCell, France). Cells were seeded in proliferation medium: NeuroCult with 10% NSC factors (StemCell), penicillin/streptomycin mix, 20 ng/ml EGF (Promega), 10 ng/ml bFGF (Promega), and cultured in suspension for 7 days at 37°C and 5% $CO_2$.

After 7 days of proliferation, neurospheres were enzymatically dissociated with Accutase (Sigma) and seeded in poly-L-ornithine-coated (Gibco) 0.17 mm glass coverslips at 100.000 cells/well. Cells were differentiated into astrocytes in serum-free medium. The following day, differentiation medium was replaced by oxygen and glucose deprivation (OGD) medium (130 mM NaCl, 5.4 mM KCl, 1.8 mM

$CaCl_2$, 26 mM $NaHCO_3$, 0.8 mM $MgCl_2$, 1.18 mM $NaH_2PO_4$, 10 mM sucrose, pH 7.2) previously saturated with $N_2$. Cells were subjected to OGD for 60 minutes in 3% $O_2$. After OGD, cells were returned to standard medium, differentiated for 7–10 days and fixed with 4% paraformaldehyde. In some experiments, hyaluronidase (OmniPur Hyaluronidase Sigma, HX0514-1) was added to the medium to remove extracellular HA. To measure metalloproteases activity in culture medium following OGD, we used the MMP Activity Assay Kit (Abcam), which employs GFP as an MMP activity reporter and DAPI as a normalization dye.

### Statistical analysis

Statistical analysis was done using GraphPad Prism software (version 8.0; GraphPad software). Data are presented as mean ± SEM unless specified otherwise. Statistical analysis was performed using unpaired Student's *t*-test for independent conditions and paired Student's *t*-test for dependent conditions. In addition, one-way and two-way ANOVA were also performed to compare more than two groups among them. Tukey and Dunnett post hoc tests were used. For HABP skeleton analysis, Kruskal–Wallis analysis was performed to compare cumulative frequency curves in more than two groups. p-Value was considered significant at $p < 0.05$. Sample number (n), p-values, and statistical test are indicated in figure legends.

### Materials availability

The AAV-Has2-NMR vector is available from the corresponding authors upon reasonable request.

## Acknowledgements

We thank Juan Carlos Chara, Zara Martínez-Paez, and Laura Escobar for their invaluable technical support. We thank Laura Simon for proofreading the manuscript, and Mónica Benito-Muñoz, and Sol Beccari for training in MCAO in mice. We also thank Vera Gorbunova (University of Rochester, USA) for providing the cDNA of HAS2 from the naked mole rat. We thank Marie-Laure Thiolat and Dr. Nathalie Dutheil for AAV production. This study received financial support from the French government in the framework of the University of Bordeaux's IdEx 'Investments for the Future' program/ GPR BRAIN_2030 (BD). FC acknowledges funding from BIOEF (BIO21/COV/012), Eukampus, Plan Complementario en Salud (PRTR-C17.I1) and the Basque Government. FNS was funded by the Spanish Research Agency (IJCI-2017-32114, PID2020-115896RJ-I00, and RYC2021-032602-I co-funded by NextGeneration EU/PRTR). CM acknowledges funding from the Basque Government (IT1203-19) and CIBERNED. AM acknowledges grant PCI2022-134986-2 funded by Spanish Research Agency /10.13039/501100011033 and European Union NextGenerationEU/PRTR. This work was also supported by the Fédération pour la Recherche sur le Cerveau, the Agence National pour la Recherche Grant ANR- 17-CE16-0025-02, ANR-21-CE16-0034-01, ANR-21-CE13-0003, and the Fondation pour la Recherche Médicale, EQU201903007806 grants to HC. MA was supported by a PhD fellowship from the Basque Government.

## Additional information

### Funding

| Funder | Grant reference number | Author |
| --- | --- | --- |
| University of Bordeaux | IdEx 'Investments for the Future' program/GPR BRAIN_2030 | Benjamin Dehay |
| Basque Government | | Fabio Cavaliere |
| Berrikuntza + Ikerketa + Osasuna Eusko Fundazioa | BIO21/COV/012 | Fabio Cavaliere |
| Eukampus | | Fabio Cavaliere |
| Ministerio de Ciencia e Innovación | PRTR-C17.I1 | Fabio Cavaliere |

| Funder | Grant reference number | Author |
|---|---|---|
| Ministerio de Ciencia e Innovación | IJCI-2017-32114 | Federico N Soria |
| Ministerio de Ciencia e Innovación | PID2020-115896RJ-I00 | Federico N Soria |
| Ministerio de Ciencia e Innovación | RYC2021-032602-I | Federico N Soria |
| Basque Government | IT-1203-19 | Carlos Matute |
| Centro de Investigación Biomédica en Red sobre Enfermedades Neurodegenerativas | | Carlos Matute |
| Basque Government | PhD fellowship | Maria Ardaya |
| Ministerio de Ciencia e Innovación | PCI2022-134986-2 | Abraham Martín |
| Fédération pour la Recherche sur le Cerveau | | Harold Cremer |
| Agence Nationale de la Recherche | ANR- 17-CE16-0025-02 | Harold Cremer |
| Agence Nationale de la Recherche | ANR-21-CE16-0034-01 | Harold Cremer |
| Agence Nationale de la Recherche | ANR-21-CE13-0003 | Harold Cremer |
| Fondation pour la Recherche Médicale | EQU201903007806 | Harold Cremer |

The funders had no role in study design, data collection and interpretation, or the decision to submit the work for publication.

## Author contributions

Maria Ardaya, Formal analysis, Investigation, Visualization, Methodology, Writing – original draft; Marie-Catherine Tiveron, Harold Cremer, Resources, Supervision, Methodology; Esther Rubio-López, Methodology; Abraham Martín, Resources, Supervision; Benjamin Dehay, Conceptualization, Resources; Fernando Pérez-Cerdá, Supervision; Carlos Matute, Funding acquisition; Federico N Soria, Conceptualization, Software, Supervision, Visualization, Methodology, Writing – original draft, Writing – review and editing; Fabio Cavaliere, Conceptualization, Formal analysis, Supervision, Funding acquisition, Writing – original draft, Project administration, Writing – review and editing

## Author ORCIDs

Maria Ardaya ⬤ https://orcid.org/0000-0002-0103-5649
Harold Cremer ⬤ https://orcid.org/0000-0002-8673-5176
Benjamin Dehay ⬤ https://orcid.org/0000-0003-1723-9045
Carlos Matute ⬤ https://orcid.org/0000-0001-8672-711X
Federico N Soria ⬤ https://orcid.org/0000-0003-1229-9663
Fabio Cavaliere ⬤ https://orcid.org/0000-0003-2003-522X

## Ethics

Animals were maintained under standard laboratory conditions with food and water ad libitum. All animals were handled in accordance with the European Communities Council Directive (2010/63/EU) and approved by the Ethics Committee of the University of the Basque Country (EHU) under license CEEA/340/2013.

Reviewer #1 (Public review): https://doi.org/10.7554/eLife.96076.4.sa1
Reviewer #2 (Public review): https://doi.org/10.7554/eLife.96076.4.sa2
Author response https://doi.org/10.7554/eLife.96076.4.sa3

## Additional files

### Supplementary files
MDAR checklist

Source data 1. Contains source data for all figures.

### Data availability
All data generated or analysed during this study are included in the manuscript and supporting files.

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
