## [Editor Report · eLife Assessment]

The authors show that a middle carotid artery occlusion (MCAO) hypoxia lesion leads to hyaluronan-mediated chemoattraction to the lesion penumbra of Thbs-4-expressing astrocytes of the subventricular zone (SVZ). These findings are **valuable** because they shed light on the function of astrocytes from the adult SVZ in pathological states like brain ischemic injury. The results are **convincing** as they rely on a comprehensive analysis of experimental data.

---

## [Referee Report · Reviewer #1 (Public review)]

Summary:

The authiors show that SVZ derived astrocytes respond to a middle carotid artery occlusion (MCAO) hypoxia lesion by secreting and modulating hyaluronan at the edge of the lesion (penumbra) and that hyaluronin is a chemoattractant to SVZ astrocytes. They use lineage tracing of SVZ cells to determine their origin. They also find that SVZ derived astrocytes express Thbs-4 but astrocytes at the MCAO-induced scar do not. Also, they demonstrate that decreased HA in the SVZ is correlated with gliogenesis. While much of the paper is descriptive/correlative they do overexpress Hyaluronan synthase 2 via viral vectors and show this is sufficient to recruit astrocytes to the injury. Interestingly, astrocytes preferred to migrate to the MCAO than to the region of overexpressed HAS2.

Strengths:

The field has largely ignored the gliogenic response of the SVZ, especially with regards to astrocytic function. These cells and especially newborn cells may provide support for regeneration. Emigrated cells from the SVZ have been shown to be neuroprotective via creating pro-survival environments, but their expression and deposition of beneficial extracellular matrix molecules is poorly understood. Therefore, this study is timely and important. The paper is very well written and flow of result logical.

Comments on revised version:

Thanks for addressing my final points.

---

## [Referee Report · Reviewer #2 (Public review)]

Summary:

In their manuscript, Ardaya et al address the impact of ischemia-induced astrogliogenesis from the adult SVZ and their effect on remodeling of the extracellular matrix (ECM) in the glial scar. The authors show that the levels of Thbs4, a marker previously identified to be expressed in astrocytes and neural stem cells (NSCs) of the SVZ, strongly increase upon ischemia. While proliferation is significantly increase shortly after ischemia, Nestin and DCX (markers for NSCs and neuroblasts, respectively) decrease and Thbs4 levels suggesting that the neurogenic program is halted and astrogenesis is enhanced. By fate-mapping, the authors show that astrocytes derive from SVZ NSCs and migrate towards the lesion. These SVZ-derived astrocytes strongly express Thbs4 and populate the border of the lesion, while local astrocytes do not express Thbs4 and localize to both scar and border. Interestingly, the Thbs4-positive astrocytes appear to represent a second wave of astrocytes accumulating at the scar, following an immediate reaction of first wave reactive gliosis by local astrocytes. Mechanistically, the study presents evidence that the degradation of hyaluronan (HA), a key component of the extracellular matrix (ECM) is downregulated in the SVZ after ischemia, potentially inducing astrogliogenesis, while HA accumulation at the lesion side represents at least one signal to recruit the newly generated astrocytes. In the aim to facilitate tissue regeneration after ischemic injury, the authors propose that the Thbs4-positive astrocytes could be a promising therapeutical target to modulate the glial scar after brain ischemia.

Strengths:

This topic is timely and important since the focus of previous studies was almost exclusively on the role of neurogenesis. The generation of adult-born astrocytes has been proven in both neurogenic niches under physiological conditions, but the implicated function in pathology has not been sufficiently addressed yet.

Weaknesses:

The study presented by Ardaya et al presents good evidence that a population of astrocytes that express Thbs4 contribute to scar formation after ischemic injury. The authors demonstrate that ischemic injury increases proliferation in the SVZ, decreases neurogenesis and increases astrogenesis. However, whether astrogenesis is a result of terminal differentiation of type B cells or their proliferation remains unclear. Here, a combination of fate mapping and thymidine analogue-tracing would have been conclusively.

---

## [Author Response]

The following is the authors’ response to the previous reviews.

**Reviewer #1 (Public review):**
Summary:The authors show that SVZ derived astrocytes respond to a middle carotid artery occlusion(MCAO) hypoxia lesion by secreting and modulating hyaluronan at the edge of the lesion (penumbra) and that hyaluronan is a chemoattractant to SVZ astrocytes. They use lineage tracing of SVZ cells to determine their origin. They also find that SVZ derived astrocytes express Thbs-4 but astrocytes at the MCAO-induced scar do not. Also, they demonstrate that decreased HA in the SVZ is correlated with gliogenesis. While much of the paper is descriptive/correlative they do overexpress Hyaluronan synthase 2 via viral vectors and show this is sufficient to recruit astrocytes to the injury. Interestingly, astrocytes preferred to migrate to the MCAO than to the region of overexpressed HAS2.Strengths:The field has largely ignored the gliogenic response of the SVZ, especially with regards to astrocytic function. These cells and especially newborn cells may provide support for regeneration. Emigrated cells from the SVZ have been shown to be neuroprotective via creating pro-survival environments, but their expression and deposition of beneficial extracellular matrix molecules is poorly understood. Therefore, this study is timely and important. The paper is very well written and the flow of results logical.Comments on revised version:The authors have addressed my points and the paper is much improved. Here are the salient remaining issues that I suggest be addressed.

We appreciate the feedback by the reviewer, and we are glad that the paper is considered to be much improved. We have done our best to address the remaining issues in this 2nd revision.

The authors have still not shown, using loss of function studies, that Hyaluronan is necessary for SVZ astrogenesis and or migration to MCAO lesions.

This is true. Unfortunately, complete removal of hyaluronan (via Hyase) triggers epilepsy, already described in 1963 by James Young (*Exp Neurol* paper). Degradation by Hyase also provokes neuroinflammation (Soria et al., 2020 Nat Commun). Two alternatives could be (1) partial depletion with Has inhibitor 4MU (but it is also associated with increased inflammation) or (2) a Has-KO mouse, such as Has3-/- (Arranz et al., 2014), although, to our knowledge, this mouse line is not openly available. We have added a sentence in line 332 addressing this shortcoming: “Loss-of-function studies, using HA-depletion models or HA synthase (Has)deficient mice are still needed to corroborate this finding, although the inflammation associated with HA deficiency might confound interpretation.”

(1) The co-expression of EGFr with Thbs4 and the literature examination is useful.

We thank the reviewer for the kind comment.

(2) Too bad they cannot explain the lack of effect of the MCAO on type C cells. The comparison with kainate-induced epilepsy in the hippocampus may or may not be relevant.

As stated in the previous response, we also found this interesting, and it does warrant further exploration by looking into possible direct NSC-astrocyte differentiation. But we believe that both this possible direct differentiation and the reactive status for these astrocytes are out of the scope of the study. We will not speculate about this in the discussion, either.

(3) Thanks for including the orthogonal confocal views in Fig S6D.(4) The statement that "BrdU+/Thbs4+ cells mostly in the dorsal area" and therefore they mostly focused on that region is strange. Figure 8 clearly shows Thbs4 staining all along the striatal SVZ. Do they mean the dorsal segment of the striatal SVZ or the subcallosal SVZ? Fig. 4b and Fig 4f clearly show the "subcallosal" area as the one analysed but other figures show the dorsal striatal region (Fig. 2a). This is important because of the well-known embryological and neurogenic differences between the regions.

While it is true that Thbs4 is also expressed in the other subregions of the SVZ (lateral, ventral and medial), as observed in Fig 8. we chose the dorsal area because it is the subregion where we observed the larger increase in slow proliferative NSCs (Thbs4/GFAP/BrdU-positive cells) after MCAO (Fig S3). As observed in the quantifications in Fig S3, we found Thbs4/GFAP/BrdUpositive cells increase in lateral, medial and ventral SVZ, but it is not significant. Therefore, from Fig 4 onwards, we focused on the dorsal SVZ, which the reviewer mentions as “subcallosal” area. We chose the term “dorsal” as stated in single-cell studies (Cebrian-Silla et al, 2021, eLife; Marcy et al., 2023, Sci Adv) and reviews (Sequerra 2014 Front Cell Neurosci) that investigate or mention this subregion, respectively. In the abstract, we are perfectly clear stating that newborn astrocytes migrate frm both dorsal and medial areas.

In Fig 2a, the immunofluorescence image shows medial and lateral SVZ, but at this point in the paper, we have not yet made specific subregional quantifications, and the Nestin, DCX and Thbs4 quantifications refer to the SVZ as a whole, both in the IF and in the WB (Fig 2e-g). We apologize for the confusion. We have clarified this in the text (line 119).

(5) It is good to know that the harsh MCAO's had already been excluded.(6) Sorry for the lack of clarity - in addition to Thbs4, I was referring to mouse versus rat Hyaluronan degradation genes (Hyal1, Hyal2 and Hyal3) and hyaluronan synthase genes (HAS1 and HAS2) in order to address the overall species differences in hyaluronan biology thus justifying the "shift" from mouse to rat. You examine these in the (weirdly positioned) Fig. 8h,i. Please add a few sentences on mouse vs rat Thbs4 and Hyaluronan relevant genes.

We thank the reviewer for these remarks. We have now added a sentence pointing to the similar internalization and degradation in rat and mouse (reviewed by Sherman et al., 2015). This correction is in line 233. Hyaluronan is, in evolutionary terms, a very “old” molecule, part of the “ancient” glycan-based matrix, before the evolution of proteoglycans and fibrous proteins such as collagen, laminin etc. Hence, its machinery is highly conserved across species.

We have also reorganized the panels in Fig 8, where 8h and 8i were indeed weirdly positioned. We hope that the new version of this figure is more easily readable.

(7) Thank you for the better justification of using the naked mole rat HA synthase.
**Reviewer #3 (Public review):**
Summary:The authors aimed to study the activation of gliogenesis and the role of newborn astrocytes in a post-ischemic scenario. Combining immunofluorescence, BrdU-tracing and genetic cellular labelling, they tracked the migration of newborn astrocytes (expressing Thbs4) and found that Thbs4-positive astrocytes modulate the extracellular matrix at the lesion border by synthesis but also degradation of hyaluronan. Their results point to a relevant function of SVZ newborn astrocytes in the modulation of the glial scar after brain ischemia. This work's major strength is the fact that it is tackling the function of SVZ newborn astrocytes, whose role is undisclosed so far.Strengths:The article is innovative, of good quality, and clearly written, with properly described Materials and Methods, data analysis and presentation. In general, the methods are designed properly to answer the main question of the authors, being a major strength. Interpretation of the data is also in general well done, with results supporting the main conclusions of this article.In this revised version, the points raised/weaknesses were clarified and discussed in the article.
**Recommendations for the authors:**

**Reviewer #1 (Recommendations for the authors):**
Minor points:(1) Thanks for the clarification.(2) Thanks for the clarification.(3) The magnification is not apparent in Fig. 5.

We had removed two brain slices (from 4 to 2) in order to increase the size of the image 2-fold. We have now further increased the TTC panel, 25% from the revised version, 125% from the original.

(4) Thanks for the clarification.(5) Thanks for the clarification.(6) Thanks for the clarification.(7) Thanks for the clarification.(8) Thanks for the clarification.